# Clinical outcomes associated with prostate cancer conspicuity on biparametric and multiparametric MRI: a protocol for a systematic review and meta-analysis of biochemical recurrence following radical prostatectomy

Naomi Morka ![ORCID],[1] Benjamin S Simpson ![ORCID],[2] Rhys Ball,[3] Alex Freeman,[3] Alex Kirkham,[4] Daniel Kelly ![ORCID],[5] Hayley C Whitaker,[2] Mark Emberton,[2,6] Joseph M Norris ![ORCID] [2,6]

ME and JMN are joint senior authors.

For numbered affiliations see end of article.

**Correspondence to**
Mr Joseph M Norris;
joseph.norris@ucl.ac.uk

## ABSTRACT

**Introduction** There is an increasing body of evidence to suggest that visibility of prostate cancer on magnetic resonance (MRI) may be related to likelihood of adverse pathological outcomes. Biochemical recurrence (BCR) after radical prostatectomy remains a significant clinical challenge and a means of predicting likelihood of this prior to surgery could inform treatment choice. It appears that MRI could be a potential candidate strategy for BCR prediction, and as such, there is a need to review extant literature on the prognostic capability of MRI. Here, we describe a protocol for a systematic review and meta-analysis of the utility of biparametric MRI (bpMRI) and multiparametric MRI (mpMRI) in predicting BCR following radical prostatectomy for prostate cancer treatment.

**Methods and analysis** PubMed, MEDLINE, Embase and Cochrane databases will be searched and screening will be guided by the Preferred Reporting Items for Systematic Reviews and Meta-analyses (PRISMA) guidelines. In order to meet the inclusion criteria, papers must be English-language articles involving patients who have had bpMRI or mpMRI for suspected prostate cancer and have undergone radical prostatectomy as definitive therapy. Patients must have had prostate-specific antigen monitoring before and after surgery. All relevant papers published from July 1977 to October 2020 will be eligible for inclusion. The Newcastle-Ottawa score will be used to determine the quality and bias of the studies. This protocol is written in-line with the PRISMA protocol 2015 checklist.

**Ethics and dissemination** There are no relevant ethical concerns. Dissemination of this protocol will be via peer-reviewed journals as well as national and international conferences.

**PROSPERO registration number** CRD42020206074.

## Strengths and limitations of this study

► This study will be the first comprehensive systematic review and meta-analysis of the utility of biparametric MRI (bpMRI) and multiparametric MRI in the prediction of biochemical recurrence following radical prostatectomy, written in-line with Preferred Reporting Items for Systematic Reviews and Meta-analyses guidelines.

► This study will employ robust statistical methods to minimise the effect of heterogeneity in the literature on the determined results.

► Heterogeneity of study cohorts may limit the strength of conclusions drawn.

► Due to limited number of studies which make use of bpMRI, there is likely to be a less extensive analysis of the prognostic utility of this modality.

► Also, due to the relatively recent introduction of prostatic MRI scans, there is limited long-term data on the clinical outcomes associated with MRI.

Radical prostatectomy is the most commonly employed definitive treatment for patients with this disease,[2] often yielding favourable oncological outcomes.[3] However, biochemical recurrence (BCR), taken to be a serum prostate specific antigen (PSA) level greater than or equal to 0.2 ng/dL on two consecutive occasions[4] occurs in approximately 15%–40% of patients.[5 6] Prognostically, BCR has high utility and is associated with increased incidence of metastatic disease.[4 7] It follows then that a reliable approach to predicting BCR could inform therapeutic decision making.[8] Several models have been proposed for prognostication of BCR following radical prostatectomy,[9–12] however, these have been

## BACKGROUND

Prostate cancer is a one of the leading causes of cancer mortality in men worldwide, with over a million new cases diagnosed in 2018 alone.[1]

criticised for apparent inaccuracy and inapplicability to certain clinical settings.[13 14] This highlights the need for a widely acceptable and accessible means of understanding prostate cancer prognosis in this patient group.

MRI has increasingly become an integral part of prostate cancer screening and diagnosis. In recent years, evidence has emerged on the utility of multiparametric MRI (mpMRI) in predicting adverse pathology with MRI-invisible tumours suggesting a good prognosis and MRI-visible tumours associated with unfavourable clinical outcomes such as extracapsular extension and seminal vesicle invasion.[15–18] Given that these pathological factors are also associated with increased likelihood of BCR,[9 10] it has been suggested that mpMRI may have a role in the prediction of BCR after radical prostatectomy.[19 20] This is also supported by studies that show that the visibility of prostate tumours on mpMRI may corelate with genetic markers of poor clinical outcome.[21] Biparametric MRI (bpMRI) has been considered as a potential alternative to mpMRI.[22] Although its diagnostic utility is still being appraised, there is a growing interest in this modality as it has been suggested that it could reduce costs and imaging times.[23 24] Further exploration of the role of MRI in predicting BCR after radical prostatectomy is necessary as its use in this way could enhance and improve existing models of prostate cancer prognosis, allowing for better clinical decision making and patient care.

The aim of this systematic review is to evaluate and collate the existing literature on the prognostication of BCR after radical prostatectomy using pretreatment bpMRI or mpMRI. This will be the first of its kind and it will assess the utility of preoperative MRI as a risk stratification tool and predictor of long-term clinical outcomes.

## METHODS AND ANALYSIS

This protocol for this systematic review has been written in compliance with the guidance set out in the Preferred Reporting Items for Systematic Reviews and Meta-analyses (PRISMA)-Protocols 2015 checklist.[25] Included studies will undergo thorough appraisal and analysis to reveal the extent to which preoperative MRI status results may be indicative of the likelihood to develop BCR after radical prostatectomy.

### Search methodology

Searches will be performed on the PubMed, MEDLINE, Embase and Cochrane databases to identify all potentially relevant studies published from July 1977 (date of first human MRI)[26] to October 2020. Free text searches and Medical Subject Headings terms will be used with suitable Boolean operators. The use of search terms such as 'prostate,' 'cancer,' 'bpMRI' and 'mpMRI,' along with alternative terms for 'BCR,' 'radical prostatectomy' and 'prognosis' will ensure a comprehensive approach. Three reviewers will be involved in article selection and evaluation and will employ the use Rayyan, a systematic review collaboration tool to improve ease of collaboration,

screening of studies and removal of duplicates. To minimise likelihood of overlooking relevant literature, reference lists for each article will be manually screened and experts in the field will be approached.

### Study selection and data extraction

Studies will be screened by three separate reviewers using their title and abstract. Following this, the full-text articles will then be scrutinised against the inclusion criteria. Any disagreements will be resolved by consensus. Exclusions will be documented with the reasons for each case, allowing generation of the PRISMA flow chart.

### Inclusion criteria

To be selected for this review, studies must involve patients with suspected prostate cancer who have undergone a pretreatment bpMRI or mpMRI, and both preradical and postradical prostatectomy serum PSA testing. Only studies which use grades of conspicuity (eg, Prostate Imaging-Reporting and Data System, PI-RADS, Likert scale, etc) to assess the relationship between MRI visibility and BCR will be included.

### Exclusion criteria

Case reports, expert opinions, conference abstracts, reviews and non-English language articles will be ineligible for inclusion. Studies with patient cohorts who have received treatment prior to surgery will also be excluded.

### Data extraction

After selection, all articles will meticulously read and examined, with relevant data points extracted and inputted onto a group spreadsheet. Three or more independent reviewers will confirm the data and check for accuracy. Collation of data will be carried out as described by our group previously.[27] The data to be collected is summarised in table 1.

### Endpoints

The main endpoint of this review will be the identification of significant differences in effect estimates such as HRs for BCR between MRI-visible (eg, PI-RADS 3–5) and MRI-invisible (eg, PI-RADS 1–2) groups. Cancer-specific and all-cause mortality will be the secondary endpoints. Moderator variables, such as MRI scoring systems (eg, PI-RADS vs Likert) and MRI scanner power will be compared, and any which are of a notable impact to the final summary estimate will be recorded.

### Risk of bias in individual studies

The bias and quality of the included studies will be assessed by three independent reviewers using the Newcastle-Ottawa score. The three sections of this system (selection, comparability and outcome) will allow for an evaluation of the methods and assessment of the quality of each study. The outcome of which will aid in the thematic synthesis. Any disagreements between the reviewers will be resolved by consensus. In the case that a study is found to be at risk of significant bias or is of unsatisfactory standard, it may

**Table 1** Data collection items

| Item no | Data title | Data type |
|---|---|---|
| 1 | Year of publication | Study characteristic |
| 2 | Author names | Study characteristic |
| 3 | Study design | Study characteristic |
| 4 | Patient population | Demographics |
| 5 | No of patients | Demographics |
| 6 | Preoperative MRI status | Demographics |
| 7 | No of years of radiologist experience | Methodology |
| 8 | MRI scoring system | Methodology |
| 9 | MRI scanner power | Methodology |
| 10 | Definition of tumour visibility | Methodology |
| 11 | Definition of BCR | Methodology |
| 12 | Total follow-up time | Methodology |
| 13 | BCR status | Outcome |
| 14 | HR of BCR | Outcome |

BCR, biochemical recurrence; ;HR, hazard ratio; MRI, magnetic resonance imaging; No, number.

be excluded. If the study is still deemed fit to be included in the analysis, this will be duly noted as a limitation in the discussion and conclusion portions of the systematic review. As previously described by us elsewhere,[28] sections of the Newcastle-Ottawa scoring system may be modified to better suit the subject area of this review and ensure a greater accuracy in the findings. Risk of bias score will be assessed as a potential moderator variable in the meta-analysis.

## Meta-analysis

HRs of BCR from univariate Cox regression analyses of MRI visibility groups will be used to create the summary estimate for the meta-analysis. Studies which have incorporated hazard ratios for MRI visibility into multivariable models will not be included in the meta-analysis unless the univariate data is available. The distribution of untransformed, logit and double-arcsine transformed ratios will be compared and will be assessed for normality using density plots and tested using the Shapiro-Wilk test. Whichever set of ratios most resemble a normal distribution will be used for further analysis. Interstudy variation will be quantified using $I^2$ and a random effects model will be fitted for estimation the summary estimate. After fitting a model to all relevant studies, leave-one-out analyses (LOO) and accompanying diagnostic plots would be used to identify influential studies including: externally studentised residuals, difference in fits values, Cook's distances, covariance ratios, LOO estimates of the amount of heterogeneity, LOO values of the test statistics for heterogeneity, hat values and weights. Each study would be removed one at a time, and the summary proportion would be re-estimated based on the remaining n-1 studies. Studies with a statistically significant influence on the fitted model would be removed as outliers and the

model refitted. Finally, a summary estimate comprising the remaining studies will be calculated to estimate the true hazard associated with MRI visibility or the inverse, an MRI invisibility. Subgroup analyses are planned between Likert and PI-RADS scoring systems. Data collection items 1, 4, 7, 8, 9, 10, 11 and 12 will be tested as potential moderators assuming a sufficient number of studies are present within each group. All data analyses and visualisation will be performed using the R statistical environment (V.3.6.1, 2019-07-05) using the 'metafor' and 'meta' packages.

## Patient and public involvement

No patients were involved in this study.

## DISCUSSION

There is an increasing need to risk stratify prostate cancers, to accurately identify disease that is more likely to take an adverse clinical course.[1 29] This systematic review aims to address this by appraising current evidence on whether tumour MRI visibility is related to BCR of prostate cancer after radical prostatectomy, therefore evaluating the potential use of bpMRI and mpMRI as risk stratification tools in this population.

Current evidence suggests an association between visibility of prostate cancer on MRI with Gleason grade and tumour size.[30 31] Furthermore, it appears that MRI conspicuity is positively correlated with D'Amico risk nomogram scores.[32] In terms of the prognostic potential of MRI, it has been demonstrated that predictive power of MRI may extend beyond its relationship with histopathological markers of disease progression. Indeed, raised PI-RADS scores[33] and apparent diffusion coefficient values have

been shown to independently predict the likelihood of BCR.[34 35]

Although there is a clear potential role for bpMRI and mpMRI in predicting BCR after radical prostatectomy, across studies, there are variations in methodology, such as biopsy sampling, definitions of MRI positivity and clinically significant prostate cancer. This, together with the unavailability of extensive long-term data on MRI outcomes may create a limitation in the ability to form robust conclusions in this review. Also, due to inexhaustive reporting of tumour characteristics, there is often a lack of data on features such as tumour volume and clinical stage within MRI-invisible and MRI-visible groups. This poses a potential difficulty in understanding the effects that these factors may have on the determined results of the meta-analysis. We hope to address the heterogeneity in the literature and encourage a more uniform approach to research in this subject area.

In summary, early evidence suggests that bpMRI and mpMRI could provide a means to predict BCR of prostate cancer after radical prostatectomy. The wide availability of MRI and its non-invasive nature makes it an ideal candidate for potential BCR risk stratification.[36] Our systematic review will assess and analyse the available evidence, with the aim of providing a holistic view of the literature. This will lead to a clearer understanding of the subject, which could inform the choice of appropriate therapy in patients with prostate cancer.

## Trial status
► Preliminary searches: started.
► Piloting of the selection study process: not started.
► Formal screening: not started.
► Data extraction: not started.
► Risk of bias assessment: not started.
► Data analysis: not started.

## Draft of search strategy for Medline, Embase, PubMed and Cochrane databases
((prostat* NOT prostatitis) AND ("cancer" OR tumo?r* OR malignancy*)) AND ("magnetic resonance imaging" OR "MRI" OR "mpMRI" OR "multi-parametric magnetic resonance imaging" OR "bpMRI" OR "bi-parametric magnetic resonance imaging") AND ("biochemical recurrence" OR "BCR" OR "biochemical failure") AND ("radical prostatectomy" OR "surgery") AND (prognos* OR predict*)

## Ethics and dissemination
There are no relevant ethical concerns. Dissemination of this protocol will be via peer-reviewed journals as well as national and international conferences.

## Author affiliations
[1]University College London Medical School, London, UK
[2]UCL Division of Surgery & Interventional Science, University College London, London, UK
[3]Department of Pathology, University College London Hospitals NHS Foundation Trust, London, UK
[4]Department of Radiology, University College London Hospitals NHS Foundation Trust, London, UK
[5]School of Healthcare Sciences, College of Biomedical and Life Sciences, Cardiff University, Cardiff, UK
[6]Department of Urology, University College London Hospital, London, UK

**Contributors** The authors' contribution includes, but is not limited to, the following: NM, BSS and JMN drafted the manuscript and created the study concept. RB, AK, AF, DK, HCW and ME provided supervision and guidance during the study. All authors reviewed and approved the manuscript in its current form. JMN is the guarantor of this work.

**Funding** Norris is funded by the Medical Research Council (MRC) (MR/S00680X/1). Simpson is funded by the Rosetrees Trust. Emberton receives research support from the UK's National Institute of Health Research (NIHR) UCLH/UCL Biochemical Research Centre.

**Competing interests** Norris receives funding from the MRC. Simpson receives funding from the Rosetrees Trust. Whitaker receives funding from the PCUK, the Urology Foundation and Rosetrees Trust. Kirkham, Freeman and Emberton have stock interest in Nuada Medical Ltd. Emberton acts as a consultant, trainer and proctor to Sonatherm Inc; Angiodynamics Inc; Exact Imaging Inc.

**Patient and public involvement** Patients and/or the public were not involved in the design, or conduct, or reporting, or dissemination plans of this research.

**Patient consent for publication** Not required.

**Provenance and peer review** Not commissioned; externally peer reviewed.

**ORCID iDs**
Naomi Morka http://orcid.org/0000-0002-1925-7223
Benjamin S Simpson http://orcid.org/0000-0003-3685-6110
Daniel Kelly http://orcid.org/0000-0002-1847-0655
Joseph M Norris http://orcid.org/0000-0003-2294-0303

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
