## [Reviewer comments · BMJ Open]

ARTICLE DETAILS

TITLE (PROVISIONAL)	Clinical outcomes associated with prostate cancer conspicuity on biparametric and multiparametric magnetic resonance imaging: a protocol for a systematic review and meta-analysis of biochemical recurrence post-radical prostatectomy.
AUTHORS	Morka, Naomi; Simpson, Benjamin Scott; Ball, Rhys; Freeman, Alex; Kirkham, Alex; Kelly, Daniel; Whitaker, Hayley; Emberton, Mark; Norris, Joseph

VERSION 1 – REVIEW

REVIEWER	Nordstrom, Tobias Karolinska Inst, Dpt Medical Epidemiology and Biostatistics
REVIEW RETURNED	15-Dec-2020

GENERAL COMMENTS	Thank you for the opportunity to review this study protocol. This review will provide important knowledge and is adequately designed and planned. I have a few minor comments that the authors might wish to consider: #1: The main endpoint is difference in BCR by "mpMRI status groups". Please consider the definition of "mpMRI status groups". -Are studies on biparametric MRI excluded? -Are only grade of conspicuity (e.g. PIRADS grade) assessed as predictor of BCR-risk? Will other assessments be included, such as visible extracapsular extension? The authors might clarify this. #2: An increasing number of publications assess MRI as a adjunct to other risk assessment tools (e.g CAPRA, see e.g. PMID 32645056) or as comparator (e.g. vs Decipher, see e.g. PMID 31566845). Please clarify how this kind of studies are treated. #3: While this is a growing and complex field of evidence, the authors might consider to provide a short description of expected limitations of this review.
---

REVIEWER	Woo, Sungmin Memorial Sloan Kettering Cancer Center
REVIEW RETURNED	02-Jan-2021

GENERAL COMMENTS	This is a protocol for a timely topic of interest in the field of prostate cancer and MRI. I only have a few minor coments: 1. Databases: excellent to include all mentioned major sources2. Reviewer number 3, sufficient
--

	3. Why does the study have to have pre-treatment PSA to be included? Of course, this would enrich data for subgroup or meta-analyses, but for the primary objective of assessing the relationship between mpMRI visibility and outcomes (BCR) lack of such information could be permitted. 4. Please provide more detail on long-term measures of outcome. Authors state "such as mortality" but should be clarified if this includes cancer-specific mortality and/or any deaths. 5. Newcastle-Ottawa scoring system suitable for quality assessment. 6. Given the "expected" heterogeneity between studies (risk groups spectrum, MRI acquisition technique [ERC used vs not], radiologist experience), it would be advised that regardless of I2 value, a random-effects model be utilized. End of comments.
--	--

VERSION 1 – AUTHOR RESPONSE

Reviewer: 1

Dr. Tobias Nordstrom, Karolinska Institute

Comments to the Author:

Thank you for the opportunity to review this study protocol. This review will provide important knowledge and is adequately designed and planned. I have a few minor comments that the authors might wish to consider:

#1: The main endpoint is difference in BCR by "mpMRI status groups". Please consider the definition of "mpMRI status groups". The authors appreciate this observation. Due to the heterogeneity in the literature in the denotation of MRI conspicuity, with some studies explicitly classifying groups as visible/invisible and others only stating that groups were classified according to scoring systems (PI-RADS >3 VS <3), we used the general term of "MRI status groups". However, as the aforementioned classification methods are all measures of conspicuity the authors will remove the phrase "mpMRI status groups" and instead use the terms "MRI-visible and MRI-invisible groups", stating PI-RADS 3-5 versus 1-2 as an example in order to avoid ambiguity.

-Are studies on biparametric MRI excluded? We thank the reviewer for this comment as including studies which use biparametric MRI will likely allow for a more robust and comprehensive analysis. We will include these studies provided that they meet the inclusion criteria. The title, main body of the manuscript text and search terms have been edited accordingly to reflect this change.

-Are only grade of conspicuity (e.g. PI-RADS grade) assessed as predictor of BCR-risk? Will other assessments be included, such as visible extracapsular extension? The authors might clarify this. Only grades of conspicuity such as PI-RADS, Likert scales, Apparent Diffusion Coefficient (ADC) and internal scoring systems for MRI visibility will be assessed as a predictor of BCR-risk. Other MRI characteristic assessments such as visible extracapsular extension and seminal vesicle invasion will not be assessed as these features are taken into account in the definition of PI-RADS 5 lesions. Also, multivariable models often adjust for these features and a comparison of multivariate hazard ratios will be included in this review. Clarifications have been made in the inclusion criteria to reflect this.

#2: An increasing number of publications assess MRI as a adjunct to other risk assessment tools (e.g. CAPRA, see e.g. PMID 32645056) or as comparator (e.g. vs Decipher, see e.g. PMID 31566845). Please clarify how this kind of studies are treated. We recognise that studies which add

MRI to existing nomograms and risk assessment tools are of significant importance and these studies will be included in the thematic synthesis provided that the inclusion criteria are met. However, these studies will not be included in the meta-analysis unless a univariate analysis of MRI prediction of BCR is provided. This is to ensure a comparative analysis between studies. This has been clarified in the meta-analysis section of the manuscript.

#3: While this is a growing and complex field of evidence, the authors might consider to provide a short description of expected limitations of this review. The authors thank the reviewer for this comment. The limitations of the review in paragraph 3 of the discussion have been expanded to address this.

Reviewer: 2

Dr. Sungmin Woo, Memorial Sloan Kettering Cancer Center

Comments to the Author:

This is a protocol for a timely topic of interest in the field of prostate cancer and MRI.

I only have a few minor comments:

1. Databases: excellent to include all mentioned major sources

The authors thank the reviewer for acknowledging our robust approach.

2. Reviewer number 3, sufficient

We thank the reviewer for this comment.

3. Why does the study have to have pre-treatment PSA to be included?

Of course, this would enrich data for subgroup or meta-analyses, but for the primary objective of assessing the relationship between mpMRI visibility and outcomes (BCR) lack of such information could be permitted. The authors agree that the pre-operative PSA values are not a necessity for the primary objective of assessing the relationship MRI and the prediction of BCR. We have removed this from the list of data collection items but will include this in analyses where possible.

4. Please provide more detail on long-term measures of outcome. Authors state "such as mortality" but should be clarified if this includes cancer-specific mortality and/or any deaths. The authors appreciate this comment and recognise the need to specify the long-term outcome measures. Cancer-specific mortality and all-cause mortality have been specified as the long term outcomes. This has been reflected in the "Endpoints" section of the manuscript.

5. Newcastle-Ottawa scoring system suitable for quality assessment. The authors appreciate the reviewer's recognition of the efforts made to ensure that high quality studies are used in this work.

6. Given the "expected" heterogeneity between studies (risk groups spectrum, MRI acquisition technique [ERC used vs not], radiologist experience), it would be advised that regardless of I² value, a random-effects model be utilized. In light of this important point, the authors will account for the baseline heterogeneity between studies by utilising a random-effects model regardless of the I² value. This change has been reflected in the meta-analysis section of manuscript.

End of comments.

Reviewer: 1

Competing interests of Reviewer: None declared

Reviewer: 2

Competing interests of Reviewer: None

VERSION 2 – REVIEW

REVIEWER	Woo, Sungmin Memorial Sloan Kettering Cancer Center
REVIEW RETURNED	18-Feb-2021
GENERAL COMMENTS	All of the previous reviewer's comments have been satisfactorily addressed. End of comments.